# Interesting Images: Endocytoscopy for In Vivo Diagnosis of Intestinal Graft-Versus-Host Disease

**DOI:** 10.3390/diagnostics15131595

**Published:** 2025-06-24

**Authors:** Timo Rath, Till Orlemann, Francesco Vitali, Abbas Agaimy, Andreas Mackensen, Markus F. Neurath

**Affiliations:** 1Department of Internal Medicine I, Division of Gastroenterology, Ludwig Demling Endoscopy Center of Excellence, Friedrich-Alexander-University Erlangen-Nurnberg, 91054 Erlangen, Germany; till.orlemann@uk-erlangen.de (T.O.); francesco.vitali@med.uni-greifswald.de (F.V.); markus.neurath@uk-erlangen.de (M.F.N.); 2Institute of Pathology, Friedrich-Alexander-University Erlangen-Nurnberg, 91054 Erlangen, Germany; 3Department of Internal Medicine V, Division of Hematology and Oncology, Friedrich-Alexander-University Erlangen-Nurnberg, 91054 Erlangen, Germany

**Keywords:** endocytoscopy, graft-versus-host disease, magnification endoscopy, endomicroscopy

## Abstract

Gastrointestinal graft-versus-host disease (GvHD) is a frequent and severe complication after allogeneic stem cell transplantation (aSCTx). Although biopsy and histopathology remain the gold standard for diagnosis of GvHD, this approach can be limited by thrombocytopenia accompanying aSCTx and the diagnostic delay associated with routine histopathology. Here, we report on two patients in which dye-based contact microscopy using a latest generation endocytoscope with 520-fold magnification enabled in vivo diagnosis of GvHD. The first patient was a 23-year-old man with acute lymphoblastic leukemia presenting with non-bloody diarrhea 3 months after aSCTx. After topical staining with crystal violet and methylene blue, endocytoscopy in the rectum showed several apoptotic epithelial cells. Histopathology confirmed GvHD grade III according to the Lerner classification. The second patient was a 59-year-old female with diarrhea 3 months after aSCTx. Apart from pathognomic apoptotic bodies, EC additionally revealed crypt lumina enlargement and mononuclear cell infiltrates in the lamina propria with subsequent crypt distension. The duration of the procedure was less than 5 min in each patient. These findings illustrate that in vivo microscopy using endocytoscopy can enable instantaneous diagnosis of GvHD with the benefit of accelerating therapeutic decisions in patients with suspected severe GvHD.

**Figure 1 diagnostics-15-01595-f001:**
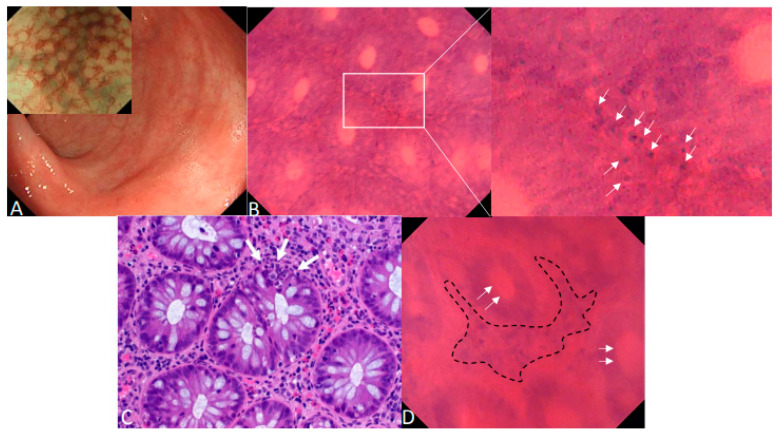
Endocytoscopy for the in vivo diagnosis of acute intestinal GvHD. Acute intestinal graft-versus-host disease (GvHD) is a frequent and life-threatening complication after allogenic stem cell transplantation (aSCTx) that targets the crypts of the gastrointestinal (GI) tract [1,2]. Histopathology with hematoxylin and eosin (H&E) staining is the widely accepted standard for diagnosis of intestinal GvHD [3,4,5] but can be limited by thrombocytopenia or, in case of co-existing liver GvHD, impaired plasmatic coagulation. Within this report, we illustrate that in vivo microscopy using endocytoscopy can enable real-time diagnosis of GvHD and therefore accelerate therapeutic decisions in patients with suspected severe GvHD. (**A**) Erythema of the mucosal surface and patchy erosions < 5 mm in the rectum under HD-WLE in a 23-year-old male patient with acute intestinal GvHD 3 months after a SCTx. NBI showed a regular vascular pattern following crypt architecture with focal hyperemia (insert picture). (**B**) After topical staining of the rectal mucosa with 1 mL of 0.5% crystal violet and 0.1% methylene blue for cytoplasmatic and cell nuclei staining, in vivo contact microscopy with 520-fold magnification with a latest generation endocytoscope (EC, Olympus GIF H290EC) showed several apoptotic cells at the basis of colonic crypts (white arrows) during ongoing endoscopy. (**C**) Corresponding standard histopathology with H&E staining confirmed GvHD grade III according the Lerner classification [6]. White arrows: apoptotic cells. (**D**) Endocytoscopy in a 59-year-old female with acute intestinal GvHD additionally revealed, apart from pathognomic apoptotic bodies, crypt lumina enlargement (white arrows) and mononuclear cell infiltrates in the lamina propria (dashed line) with subsequent crypt distension. Similar to the first patient, GvHD was confirmed by H&E staining. Previous studies have successfully utilized confocal laser endomicroscopy (CLE) for the in vivo diagnosis of acute intestinal GvHD [7]. However, single cell discrimination is limited with CLE and, therefore, the visualization of apoptotic cells as the pathognomic hallmark of GvHD might be facilitated with EC since both cytoplasmatic and cellular stainings are topically administered during the procedure. Larger studies are required to confirm that contact microscopy with EC can enable in vivo diagnosis of GvHD and therefore can accelerate therapeutic decisions (e.g., intensification of immunosuppressive therapy) in patients with suspected severe GvHD.

## Data Availability

The original contributions presented in this study are included in the article. Further inquiries can be directed to the corresponding author.

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
