# Peer review of "Interesting Images: Endocytoscopy for In Vivo Diagnosis of Intestinal Graft-Versus-Host Disease"

_diagnostics, 2025, doi:10.3390/diagnostics15131595_

Round 1
Reviewer 1 Report
Comments and Suggestions for Authors
The article presents a visually and clinically significant case-based report on the use of high-resolution endocytoscopy for the real-time in vivo diagnosis of gastrointestinal graft-versus-host disease.
The manuscript is succinct and well-structured for its intended format (short report in "Interesting Images").
Only two cases are presented, limiting generalizability. While suitable for an image-based report, the broader clinical value would benefit from more cases or preliminary comparative data.
For a broader audience, integrating a brief comparison with other endoscopic modalities and a short mention of limitations or future directions would enhance impact.
Please review the article from a grammar (e.g., "conflicts of interests" should be "conflict of interest") , but also from a technical editing point of view (title with the same font, lack of punctuation marks).
Author Response
Reviewer 1 (R1):
R1: The article presents a visually and clinically significant case-based report on the use of high-resolution endocytoscopy for the real-time in vivo diagnosis of gastrointestinal graft-versus-host disease.
The manuscript is succinct and well-structured for its intended format (short report in "Interesting Images").
Author’s reply: We thank the reviewer for these kind words and for describing our manuscript as succinct and well-structured.
R1: Only two cases are presented, limiting generalizability. While suitable for an image-based report, the broader clinical value would benefit from more cases or preliminary comparative data.
For a broader audience, integrating a brief comparison with other endoscopic modalities and a short mention of limitations or future directions would enhance impact.
Author’s reply: We agree that, based on these two cases presented, generalizability is limited but are glad to hear that the reviewer considers our manuscript as suitable for an image-based report. As suggested by the reviewer, we included a brief comparison with other endoscopic modalities and a brief outlook on future directions.
R1: Please review the article from a grammar (e.g., "conflicts of interests" should be "conflict of interest") ,but also from a technical editing point of view (title with the same font, lack of punctuation marks).
Author’s reply: We apologize for these oversights and corrected grammar and technical aspects accordingly.
Reviewer 2 Report
Comments and Suggestions for Authors
A good study. It can provide significant contributions to the literature. However, some revisions need to be made.
1-In the Abstract section, a brief sentence should indicate how long it took to obtain the results of the procedure performed for the diagnosis of these two patients.
2-Introduction, Case-1, Case-2, Discussion, Conclusion sections should be added.
3-The demographic information of the cases should be provided in a table.
4-In the conclusion section; it should be explained what benefits this procedure brings to the patients.
Author Response
Reviewer 2 (R2):
R2: A good study. It can provide significant contributions to the literature. However, some revisions need to be made.
Author’s reply: We thank the reviewer for describing our work as a good study and for stating that it can provide significant contributions to the literature
R1: In the Abstract section, a brief sentence should indicate how long it took to obtain the results of the procedure performed for the diagnosis of these two patients.
Author’s reply: Great point! We included a sentence indicating the duration of the procedure as suggested by the reviewer.
R2: Introduction, Case-1, Case-2, Discussion, Conclusion sections should be added.The demographic information of the cases should be provided in a table.
Author’s reply: We thank the reviewer for the idea to structure our case reports with Introduction, Case presentation, Discussion and Conclusion. However, the manuscript was required to be submitted to the category “Interesting Images” by the Journal, as this manuscript category does not allow a structure with Introduction, Discussion and Conclusion, unfortunately.
R2: In the conclusion section; it should be explained what benefits this procedure brings to the patients.
Author’s reply: Excellent Point! Timely diagnosis of acute GvHD is of central importance in order to intensify immunosuppressive therapy and while histopathology with hematoxylin and eosin (H&E) staining is currently regarded as the gold standard for diagnosis of intestinal GvHD, obtaining biopsies can be limited by the presence of thrombocytopenia. Therefore, replacement of coagulation factors and platelets is often necessary before biopsies can be taken. Furthermore, and potentially more relevant, standard histopathology is frequently associated with a certain diagnostic delay and therefore might limit timely diagnosis of intestinal GvHD. Therefore, Endocytoscopy holds the potential (i) to make instantaneous diagnosis of GvHD during endoscopy and (ii) to accelerate therapeutic decisions (e.g. intensification of immunosuppressive therapy).
Due to space limitations in the “Interesting Image” category, under which our report was submitted, we cannot fully discuss these aspects, but re-verbalized our manuscript accordingly to strengthen the aspect of instantaneous diagnosis and decision-making and thank the reviewer for this excellent point.
Round 2
Reviewer 2 Report
Comments and Suggestions for Authors
The authors fulfilled their responsibilities. The work can be published in this form.